# Axiomatisation and Simulation

**Klaus G. Troitzsch** [†] 

Institut für Wirtschafts- und Verwaltungsinformatik, Universität Koblenz-Landau, Postfach 201602, 56016 Koblenz, Germany; klaus.g.troitzsch@bluewin.ch; Tel.: +49-171-3226462

† Current address: Schillerstraße 26, 79713 Bad Säckingen, Germany.

**Abstract:** The paper discusses the relation between the "non-statement view" of the structuralist program in philosophy of science and agent-based simulation and the use of this relation for a deeper understanding of the verification and the validation of simulation models. To this end it uses the history of the gender desegregation process in German schools in the second half of the 20th century and two simulation models trying to explain and understand this historical process. The relation between the two simulation models on one hand and the structuralist reconstruction of the mental and verbal theory of the observed phenomenon is depicted step by step, showing the verification of the more recent simulation model along the lines of the formal definition of this theory. Finally, the simulation model is used to make two unobservable parameters measurable with the help of the formalised theory, which allows new insights into the historical process.

**Keywords:** non-statement view; "third way of doing science"; translating simulation models; model verification; gender segregation

## 1. Introduction

In contrast to earlier publications on the same topic ([1], pp. 124–177), [2–6], this paper aims at a logical reconstruction of a theory trying to explain the process of overcoming gender segregation with an intended empirical application to a history of four decades of employing male and female teachers in schools in the German federal state of Rhineland-Palatinate where a research project in the 1990s produced sufficient empirical data. The data and its analysis were first published in [7,8] and later on used in ([9], pp. 113–119) as well as in ([10], pp. 415–420). Beside the discussion of an interesting example of some complexity, the paper mainly serves as an introduction to logical reconstructions of theories (The term "logical reconstruction" is used here in the same sense as, e.g., in [11], one of the authors of [12], and of many other papers about "semantic reconstruction" or "structuralist reconstruction", see [13]) in the social sciences—a task that has not been performed so very often [14]. This use case is particularly appropriate for three reasons: (1) it sufficiently complex, with three levels of agents (ministry, school, teachers), (2) both a fairly long time series of empirical data is available for the two upper-level agents and the distribution of one attribute of the lowest-level agents is available for forty years, and (3) an obsolete simulation model is available for which a method of replication can be shown which can be adapted to other obsolete models.

Moreover, this paper uses the example as a means of discussing the role of simulation in doing science, refuting the idea that simulation is a "third way of doing science" [15] or "a distinctively new kind of scientific method. intermediate in kind between empirical experimentation and analytic theory ([16], p. 103). This is done in Section 2 where the use of simulation or numerical experimentation

is classified as the use of a third symbol system [17] beyond natural language and "analytical mathematical technique" ([16], p. 104). For a classification of the "ways of doing science" and the symbol systems used in science we shall use a diagram (Figure 1) showing various connotations of "theory", "model" and "target system" with the scientific practices connecting them.

Sections 3 and 4 discuss the model which we use as an example and describe the translation between the original gender desegregation simulation model used in [7,9] and a reconstruction along the lines of the non-statement view of the structuralist program [12] as well as the translation between such a reconstruction (which we will call **GDS**) and a new simulation model programmed in a more recent theory-based simulation environment, namely NetLogo [18]. The step taken in Section 4 supports the verification of the model, as the correspondence between a formal description of the (*models* of a) theory and the resulting simulation model becomes very transparent.

Section 5 shows the results of the (new) simulation model, a sensitivity analysis, an approach at validation and, finally, its ability to produce an estimate of historical values of a **GDS**-theoretical term. The conclusion, finally, tries to encourage the use of the kind of formalisation applied in this paper.

## 2. The Role of Simulation in the Process of Theory Building

Figure 1, slightly redrawn from [19], shows an expanded framework to extend the logic of the basic model of the modelling and simulation process in ([9], p. 17) which only contained the items and arrows marked in red in the new diagram. The "*Abstraction*" arrow (now dotted) connecting the "*target system*" (the real world issue in Figure 1) to the "*model*" (the executable simulation model in the new diagram) was replaced by a chain of arrows (now in green) from the real world issue via a mental model, a *potential model* and a *full model* in the sense of the non-statement view of the structuralist program [12] and a theory-based simulation environment to the executable simulation model.

The words in italics between quotation marks are the terms used in ([9], p. 17); in the rest of the paper "*model*" in italics refers to the usage of the word in the non-statement view of the structuralist program, otherwise "model" refers to the usage in the context of "modelling and simulation", "agent-based modelling" etc., always taking into account that a "simulation model" is likely to be an element of the set of *full models* of some theory, even if the simulation modeller has not make this explicit.

The three levels of Figure 1 refer to the most complete information in the middle ("full model", executable simulation, and real world target system, although part of the latter is usually hidden), the observable aspects of model and target system on the top level, and the assumption of what is the case on the bottom level.

This framework is meant to not only provide future simulation researchers with a research architecture that can assist them in their modelling efforts, but also help the wider research community put simulation methods into perspective. The following subsections explain the different building blocks of the research architecture and its relationships—with a column containing various aspects of "desk research", i.e., the literature overview with which one usually starts, at the right end, a column with several aspects of empirical research and two columns with several stages of theoretical research with and without simulation at the left end of the diagram, where these two columns symbolise the two symbol systems [17] appropriate for a formal treatment of theoretical problems (as opposed to natural language which is often ambiguous). When Axelrod [15] called theory building with simulation the "third way of doing science" as "contrasted with the two standard methods of induction and deduction", this paper argues that both ways—with more 'classical' methods of induction and deduction and with computer simulation—are the same way of doing science, the more so, as more often than not—and the more, the more complex the target systems are—interesting mathematical models have no closed solutions and have to be solved with the help of numerical methods, and when Gilbert and Terna [20] called social simulation "a 'third way' of carrying out social science, in addition to argumentation and formalisation, they justly distinguished it from verbal argumentation (Ostrom's first symbol system), but a simulation program is as highly formalised as any mathematical formalisation.

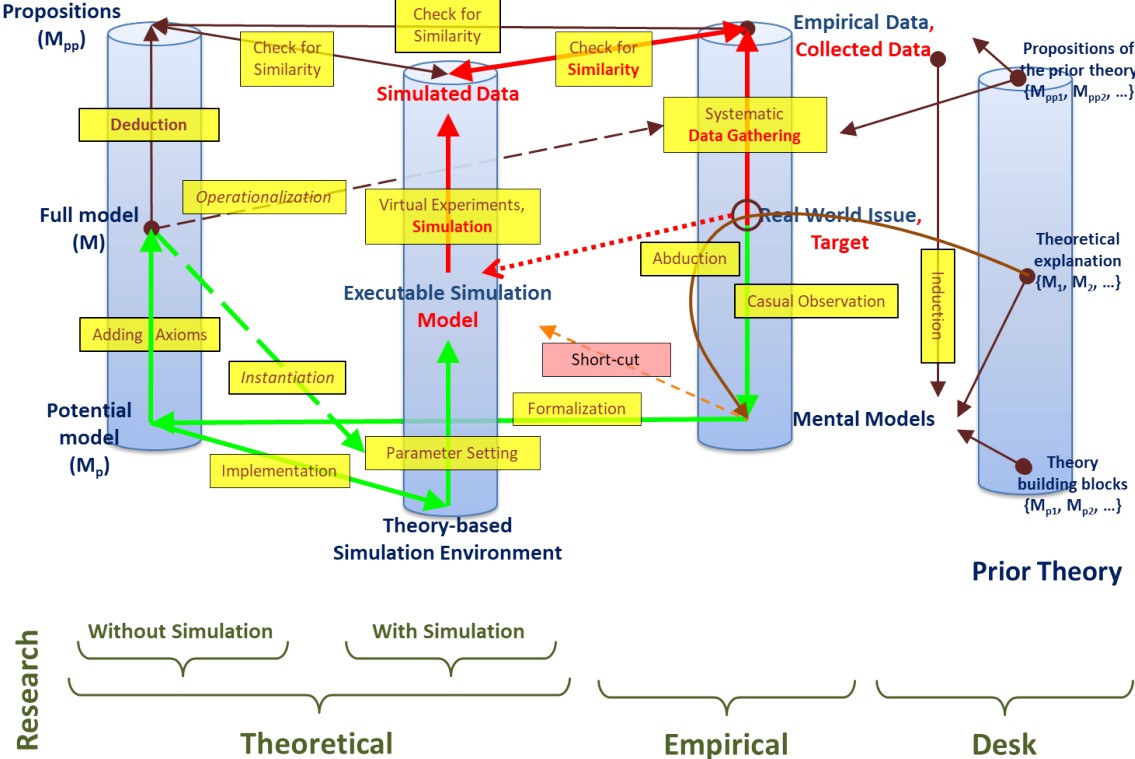

**Figure 1.** A new research architecture.

## 2.1. Prior Theory—Desk Research

Scientists usually do not start from scratch when they attempt to explain some empirical phenomenon with some theory. They rather start from the existing literature dealing with the empirical phenomenon under discussion. This is sometimes called "desk research", a term mainly used by market researchers whereas academics prefer the term "literature review" ([21], p. 175) defined by [22] as "a systematic, explicit and reproducible method for identifying, evaluating and interpreting the existing body of recorded work produced by researchers, scholars and practitioners." In the framework of Figure 1 desk research serves mainly two purposes: finding research literature on the problem in question and finding a potential target system from which data about the problem in question is available.

In the current context, this meant an in-depth literature analysis [8] and the retrieval of historical data about gender relations (both for teachers and pupils) in schools in a region for which such data were available. In terms of [22] the data finally retrieved came from different sources (school archives, ministry archives etc), i.e., they were not ready-made by earlier researchers but at best by practitioners, such that these data can rather be qualified as data generated by empirical research than as data prepared for secondary analysis ([23], p. 33).

From the literature one first axiom soon arose (from "prior theory" in terms of Figure 1) all the way along the green arrows up to the "full model", namely that all teachers leaving their jobs had to be replaced by men and women with equal overall probability, following the equal opportunities principle already laid down in article 3 linea 2 of the German Basic Law of 1949 (although ordinary legislation procrastinated the equal opportunities for women for many years), another assumption was that men stay in their jobs longer than women do (an assumption which was later on corroborated and precised mainly in stakeholder interviews as quantitative empirical data for the period under observation were hard to come by). As a result of the popular tradition to teach boys and girls in separate schools (run mainly by teachers of the same sex as the pupils with the typical exception of

male math and physics teachers in girls' schools) it was also assumed that female applicants would have had a higher probability to be allocated to schools where a majority was female.

*2.2. Empirical Research*

Beside the historical school data, empirical research was done in form of interviews with witnesses (officials in the ministry, school headmasters, members of parents' councils) to shed some light on the employment decisions made during the four decades under observation. Changes in gender relations turned out to be partially caused by more and more girls wanting to attend schools of the highest grade (or more and girls' parents wanting their daughters to attain better qualified education). As traditionally in Germany schools of the highest grade ("Gymnasien") were not co-educational schools, the small number of higher girls' schools soon ceased to be sufficient for the increasing demand, and in less densely populated parts of the country the most economical response was to accept girls in schools that until then had been reserved for boys.

*2.3. Theory Building*

Formal theory building began when the verbal assumptions mentioned in the previous two subsections had to be translated into some mathematical or computational structure to replicate the time series generated from the historical data. The main purpose of this translation was to make the mentioned assumptions as least plausible (or to refute them) in the light of the empirical data. With 114 schools and four time series per school (number of male and female teachers and of girls and boys per school year) it soon became clear that the usual statistical techniques would not suffice to refute or to corroborate the assumptions. This was the origin of the MIMOSE (MIMOSE [24,25] is a meanwhile obsolete simulation language which was designed for multilevel simulation.) simulation model ([9], pp. 113–119) which replicated graphics of the frequency distribution of the gender ratios of teachers derived from the historical data to a high degree of visually perceptible similarity. But the step from the mental model of the real world issue derived from desk research and empirical research was—at the time of its first appearance—done along the dashed orange arrow labelled "Short-cut" in Figure 1. It might have been desirable to translate the mental model with much more formal methods from the very beginning but the old MIMOSE model might be a good start into a discussion how such a formal translation could have been done:

**define the base sets:** Which active and passive elements constitute the system under observation (the real world issue)?

**define the features:** Which properties do the active and passive elements of the system have which can account for the observed process?

**define the dependencies:** How do the properties of the elements depend on each other, how do they interact with each other?

The fact that the simulation model was not written in one of the general purpose languages (like PASCAL or C at that time), but in MIMOSE, a functional and partly object oriented language with object types for base sets, their instance attributes for the features and functions, but no procedures for the dependencies, certainly led to formal answers to the questions listed above; hence, it might be appropriate to depict the process along the lines of a deconstruction of the MIMOSE model and, from its results, of a new construction of a simulation model using a more up-to-date toolbox.

## 3. From a Simulation Model to a Structuralist Reconstruction

In an earlier paper [3] we showed that the formalization of a theory in a MIMOSE [24–26] simulation program is equivalent to a structuralist reconstruction of this theory—which should not come as a surprise since the design of MIMOSE (a functional, declarative language which strictly separated model description, model initialisation and simulation setup) was partly guided by the thoughts of the founders of the structuralist program. And this equivalence also holds true for

the gender desegregation model (in both versions, MIMOSE and NetLogo) and the structuralist reconstruction attempted in this section. Hence there is nothing special about computer simulation as a way of doing science, as the same induction from the historical data and the deduction from some first principles would have been (and was, in [8,27]) in a traditional approach to explain and understand the process of overcoming the gender segregation in these German schools.

According to [12] a formal definition of a theory that could explain what happened to teachers in the gymnasien of Rhineland-Palatinate—a **g**ender **des**egregation theory—starts with the definition of a theory element $\mathbf{GDS} = \langle \mathbf{K}, \mathbf{I} \rangle$ where $\mathbf{K}$ is called the theory-core and $\mathbf{I}$ is the domain of intended applications of $\mathbf{GDS}$ ([12], p. 39), one of these intended applications being the history of the employment of male and female teachers in these schools between 1950 and 1990 (other intended applications could be histories of teacher employment in schools in other parts of Germany or of countries which traditionally separated girls and boys in secondary education). Generally speaking, the theory-core is "an entity constituted by five different components: a class $\mathbf{M}_p$ (the class of *potential models*), a class $\mathbf{M}$ of *(actual) models*, a class $\mathbf{M}_{pp}$ of *partial potential models* relative to $\mathbf{M}_p$ and $\mathbf{M}$, a global constraint $\mathbf{GC}$ and a global intertheoretical link $\mathbf{GL}$ ([12], p. 79, my italics):

$$\mathbf{K(GDS)} = \langle \mathbf{M}_p\mathbf{(GDS)}, \mathbf{M(GDS)}, \mathbf{M}_{pp}\mathbf{(GDS)}, \mathbf{GC(GDS)}, \mathbf{GL(GDS)} \rangle \tag{1}$$

The set of *potential models* $\mathbf{M}_p\mathbf{(GDS)}$ is defined as a list of all terms involved; the set of *actual models* $\mathbf{M(GDS)}$ is a subset of $\mathbf{M}_p\mathbf{(GDS)}$ containing all those *models* which fulfil the axioms of the theory, and the set of the *partial potential models* $\mathbf{M}_{pp}\mathbf{(GDS)}$ is derived from $\mathbf{M}_p\mathbf{(GDS)}$ by omitting all terms whose meaning is only assigned by the theory, i.e., those terms which can only be measured with the help of the theory; $\mathbf{M}_{pp}\mathbf{(GDS)}$ may only contain terms measurable with the help of other theories $\mathbf{T'}$ linked to $\mathbf{GDS}$ by intertheoretical links collected in $\mathbf{GL}$. In our case the constraint is of little relevance, as it refers to relations between distinct *potential models*—one could think here of the history of teacher employment in Rhineland-Palatinate and other neighbouring federal states influencing each other in some way or other; another constraint could refer to another application to the same history with information on the individual successions between teachers leaving a certain school and his or her successor, as from all these successions the parameters $\kappa$ and $\delta$ could be directly estimated—for such a constraint in an entirely different context see ([2], p. 73, 84).

To perform the transformation from a simulation program into a structuralist reconstruction the following steps were identified to be necessary to achieve a definition of the set of *potential models* of a theory (Taken from [3], pp. 174–175)—and these steps will now be taken for the MIMOSE model published in [9]:

- Take the object types of the MIMOSE model as base sets of the definition of the *potential model*.

  Table 1 shows the correspondence between the object types and the base sets.

**Table 1.** MIMOSE types and base sets.

| MIMOSE Type | Base Set |
|---|---|
| system | $\mathcal{M}$, which is a singleton representing the whole state or its responsible ministry |
| schooltype | $\mathcal{G}$, a finite set whose elements represent three different kinds of gymnasium, those reserved for girls, those reserved for boys and coeducative schools [a] |
| school | $\mathcal{S}$, a finite set whose elements represent the individual gymnasia of the state in question |
| teacher | $\mathcal{P}$, a finite set whose elements represent the individual teaching persons employed by the state in question, teaching in the gymnasia of the state |

[a] In Germany and some other Continental countries, this is a school of the highest grade, preparing pupils for universities [28].

- Take $T$ as a set of instants and—if any random functions are applied in the MIMOSE program—$\langle \Omega, \mathcal{F}, P \rangle$ as a probability space to be additional base sets of the definition of a *potential model*.
- Take any constant attribute of any MIMOSE object type as a function from this object type to the attribute type (for the sake of simplicity, we identify the "type" with the "set of its instances").

  Table 2 shows the correspondence between the constant attributes of each MIMOSE type and the respective function from this object type to the respective attribute type.

**Table 2.** Constant attributes of the MIMOSE types and their corresponding function signatures.

| MIMOSE Type Attribute and Type | Function Signature |
|---|---|
| `system` `schooltypes :  list of schooltypes` | $\gamma : \mathcal{M} \to \mathbb{P}(\mathcal{G})$ |
| `schooltype` `schools :  list of schools` | $\sigma : \mathcal{G} \to \mathbb{P}(\mathcal{S})$ |
| `school` | (no constant attributes) |
| `teacher` `position :  list of school` `sex :  int` | $\phi : \mathcal{P} \to \mathcal{S}$ $\beta : \mathcal{P} \to \{0, 1\}$ |

There are two more global terms (constants whose values were set in the initialisation phase) in the original MIMOSE program, namely $\nu$ and $\kappa$ and one more global term first introduced in [10], namely $\delta$ which need to be mentioned in the definition of the *potential model* of **GDS**.

- Take any variable attribute of any MIMOSE object type as a function from the cross product of object type and the set of instants $\mathcal{T}$ to the attribute type.

  Table 3 shows the correspondence between the variable attributes of each MIMOSE type and the respective function from this object type to the respective attribute type.

**Table 3.** Variable attributes of the MIMOSE types and their corresponding function signatures.

| MIMOSE Type Attribute and Type | Function Signature |
|---|---|
| `school` `SR : real` [a] `SRp :  real` [b] `lteacher :  list of teacher` `sexRatio :  real` `sexRatioList :  list of real` `prob1 :  real` | $\mathbb{P}(\mathcal{S}) \times \mathcal{T} \to \mathbb{R}$ $\mathbb{P}(\mathcal{S}) \times \mathcal{T} \to \mathbb{R}$ $\tau : \mathcal{S} \times \mathcal{T} \to \mathbb{P}(\mathcal{T})$ $x : \mathcal{S} \times \mathcal{T} \to \mathbb{R}$ $\lambda : \mathcal{S} \times \mathcal{T} \to \mathbb{P}(\mathbb{R})$ $\pi : \mathbb{R} \times \mathcal{T} \to [0, 1)$ |
| `teacher` `age :  int` `duration :  int` `cond :  int` `death :  list of teacher` [c] `new :  list of teacher` [c] | $\alpha : \mathcal{P} \times \mathcal{T} \to \{25, .., 65\}$ $\rho : \mathcal{P} \times \mathcal{T} \to \{0, .., 40\}$ $\omega : \{25, .., 65\} \times \{0, .., 40\} \to \{0, 1\}$ $\epsilon : \mathcal{S} \times \mathcal{T} \to \mathbb{P}(\mathcal{T})$ $\mu : \mathcal{S} \times \mathcal{T} \to \mathbb{P}(\mathcal{T})$ |

[a] Treated as an auxiliary variable, see text below. [b] Treated as an auxiliary variable, see text below. [c] In the MIMOSE program, the retiring and replacing `teacher` operated on the `lteacher:  list of teachers` of its `school`, deleting and appending itself, respectively. This is why the respective functions in the definition of the *potential model* have sets as their ranges.

- After these first four steps the definition of the *potential model* is complete.

- Take any function application ( in MIMOSE, every variable attribute had such a function application bound to it as part of the object type definition) as an axiom of the definition of the *model*. This completes the definition of the *model* of the theory.

Finally, this leads to the definition of a *potential model* $\mathbf{M}_p(\textbf{GDS})$ of the **g**ender **des**egregation theory **GDS**:

$\mathbf{M}_p(\textbf{GDS})$: $\gamma$ is a *potential model* of the gender desegregation theory ($\gamma \in \mathbf{M}_p(\textbf{GDS})$) iff there exist $\mathcal{T}, \langle \Omega, \mathcal{F}, P \rangle, \mathcal{M}, \mathcal{G}, \mathcal{S}, \mathcal{P}, \delta, \kappa, \nu, \sigma, \tau, x, \lambda, \pi, \phi, \beta, \alpha, \rho, \omega, \epsilon, \mu$ such that

1. $\gamma = \langle \mathcal{T}, \langle \Omega, \mathcal{F}, P \rangle, \mathcal{M}, \mathcal{G}, \mathcal{S}, \mathcal{P}, \delta, \kappa, \nu, \sigma, \tau, x, \lambda, \pi, \phi, \beta, \alpha, \rho, \omega, \epsilon, \mu \rangle$;
2. $\mathcal{T} \to \mathbb{N}$ is a bijective function [labeling (or coordinatizing) the instants ($t \in T$)—start of a school year—with integer numbers ($\in \mathbb{N}^+$)],
3. $\langle \Omega, \mathcal{F}, P \rangle$ is a probability space with

    (a) $\Omega$ is a sample space,
    (b) $w : \mathcal{S} \to \Omega$ is bijective,
    (c) $\mathcal{F} \subseteq \Omega$ is a family of events,
    (d) $P$ is a probability measure defined on $\mathcal{F}$.
4. $\langle A, \mathcal{A} \rangle$ is a measurable space [of lots drawn to select new teachers] with $A = [0.0, 1.0)$,
5. $\mathcal{M}$ is a singleton [ministry],
6. $\mathcal{G} = \{f, b, c\}$ is a set [containing the types schools may belong to: girls', boys' and coeducative schools],
7. $\mathcal{S}$ is a finite set [of schools],
8. $\mathcal{P}$ is a finite set [of teaching persons],
9. $\kappa$ is a real-valued constant [this parameter of the function $\pi$ determines the strength of the influence of the current gender relation in a certain school on the employment of another female or male teacher],
10. $\delta \in \mathbb{R}^+$ is a positive real-valued constant [this parameter of the function $\pi$ determines the willingness of the employing ministry to use an equal-opportunities policy; $\delta < 1$ favours men, $\delta > 1$ favours women],
11. $\nu$ is a function with $\text{Dom}(\nu) = \mathcal{T}$ and $\text{Rge}(\nu) = \mathbb{R}$ [$\nu(t)$ yields a value that makes sure that the countrywide probability of newly employing a man or a woman is kept constant at all times—this probability is 0.5 if $\delta = 1.0$, for $\delta < 1$ the probability to employ a woman is $< 0.5$, for $\delta > 1$ this probability is $> 0.5$],
12. $\sigma$ is a function with $\text{Dom}(\sigma) = \mathcal{M} \times \mathcal{G}$ and $\text{Rge}(\sigma) = \mathbb{P}(\mathcal{S})$ [$\sigma(m, g)$ yields the set of schools of type $g \in \mathcal{G}$ under the control of ministry $m \in \mathcal{M}$],
13. $\tau$ is function with $\text{Dom}(\tau) = \mathcal{S} \times \mathcal{T}$ and $\text{Rge}(\tau) = \mathbb{P}(\mathcal{P})$ [$\tau(s, t)$ yields the set of teaching persons employed at school $s$ during school year $t$],
14. $x$ is a function with $\text{Dom}(x) = \mathcal{S} \times \mathcal{T}$ and $\text{Rge}(x) = [0, 1]$ [$x(s, t)$ yields the proportion of women teaching at school $s$ at time $t$],
15. $\lambda$ is a function with $\text{Dom}(\lambda) = \mathcal{S}$ and $\text{Rge}(\lambda) = \mathbb{P}(\mathbb{R})$ [$\lambda(s)$ yields the time series of the proportion of women teaching at school $s$ for each of the school years],
16. $\pi$ is a function with $\text{Dom}(\pi) = \mathcal{S} \times \mathcal{T}$ and $\text{Rge}(\pi) = [0, 1)$ [$\pi(s, t)$ yields the probability that that a women will replace a retired teaching person or fill a newly created position at school $s$ at time $t$],
17. $(r_{t,p})_{t \in \mathcal{T}, p \in \mathcal{P}}$ is a stochastic process on the probability space $\langle \Omega, \mathcal{F}, P \rangle$ with values in the measurable space $\langle A, \mathcal{A} \rangle$ such that $r$ is univariate uniform white noise [i.e. $r(t, p)$ yields the number to be compared to $\pi(s, t)$ such that a woman is employed for $\pi(s.t) < r(t)$ at any point $t$ of time and for any candidate $p$],
18. $\phi$ is a function with $\text{Dom}(\phi) = \mathcal{P}$ and $\text{Rge}(\phi) = \mathcal{S}$ [$\phi(p)$ yields the school $s$ which employs teacher $p$],
19. $\beta$ is a function with $\text{Dom}(\beta) = \mathcal{P}$ and $\text{Rge}(\beta) = \{0, 1\}$ [$\beta(p)$ yields a proxy for the gender of teaching person $p$, where 1 means female and 0 means male],
20. $\alpha$ is a function with $\text{Dom}(\alpha) = \mathcal{P} \times \mathcal{T}$ and $\text{Rge}(\alpha) = \{25, \dots, 65\}$ [$\alpha(p, t)$ yields the age of teaching person $p$ at the start of the school year $t$],
21. $\rho$ is a function with $\text{Dom}(\delta) = \mathcal{P} \times \mathcal{T}$ and $\text{Rge}(\delta) = \{0, \dots, 40\}$ [$\rho(p, t)$ yields the number of years until a person retires, counted from the start of school year $t$],

22.	$\omega$ is a function with $\text{Dom}(\omega) = \mathcal{P} \times \mathcal{T}$ and $\text{Rge}(\omega) = \{0, 1\}$ [$\omega(p, t)$ is 1 if teacher $p$ will retire at the end of school year $t$ and 0 otherwise],

23.	$\epsilon$ is a function with $\text{Dom}(\epsilon) = \mathcal{P} \times \mathcal{T} \times \mathbb{P}(\mathcal{P})$ and $\text{Rge}(\epsilon) = \mathbb{P}(\mathcal{P})$ [$\epsilon(p, t, P)$ yields the set $P$ of teachers employed at school $s = \phi(p)$ during school year $t$ without teacher $p$ who retires at the end of this year],

24.	$\mu$ is a function with $\text{Dom}(\nu) = \mathcal{P} \times \mathcal{T} \times \mathbb{P}(\mathcal{P})$ and $\text{Rge}(\nu) = \mathbb{P}(\mathcal{P})$ [$\mu(p, t, P)$ yields the union of the set of teachers employed at school $s = \phi(p)$ at the start of school year $t$, i.e., without those who retired at the end of the previous year, with the set which contains only the newly employed teaching person $p$].

25.	$f$ is a function with $\text{Dom}(f) = \mathbb{R} \times \mathcal{T}$ and $\text{Rge}(f) = \mathbb{R}^+$ [$f(x; t)$ is the probability density function of the distribution of the proportion of women teaching at school $s \in \mathcal{S}$, and $F : \mathbb{R} \rightarrow [0, 1]$ is the corresponding cumulative density function of this proportion.

Some of these terms gain their meaning only by virtue of **GDS**, as without **GDS** they cannot be measured. The base sets (including the additional base set $\mathcal{T}$) are easily measured in real world settings, as states and their ministries, schools and their school types as well teachers are easily identified with their names. This also holds for the teaching staff of each school and the gender relation (including the history), the gender and age of all teachers at all times and the number of years until they will retire (not even excluding cases when teachers leave their jobs prematurely, which is not known in advance but anyway at the time when they announce that they want to leave). The probability density function $f$ and the corresponding cumulative density function $F$ are also observable without supposing **GDS**, as these can be easily calculated as frequencies and cumulative frequencies from the data known for all schools and all school years.

Hence what remains unmeasurable without the help of **GDS** which first introduces these terms is the function

$$\pi(s, t) = \nu(t)\delta \exp[\kappa x(s, t)] \tag{2}$$

and its coefficients which yields the probability that a woman will replace her predecessor or fill a newly created position at school $s$ at the beginning of school year $t$. According to the assumptions that went into **GDS** this probability depends on three parameters two of which are (in a first stage of **GDS**) understood as constant. The exponential function was used in analogy to earlier work by [29] as it is—unlike many other increasing functions—allows for the bimodal probability distributions which had been observed in the empirical data.

- $\delta$ is 1, granting equal opportunities to men and women or $\neq 1$ violating equal opportunities.
- $\kappa$ is used to variate the strength of the influence of an equal opportunities strategy on the gender desegregation process in the individual schools—$\kappa = 0$ means that schools dominated by female teachers or dominated by male teachers will continue to be like this for a long time, whereas $\kappa \gg 1$ means that before long all schools will have the same desired gender relation (determined by $\delta$).

For calculating $\nu$ three auxiliary terms are necessary (note that $\nu$ is calculated in the old school year but applied in the next school year).

- $\tau_\epsilon(t) : \mathcal{T} \times \mathcal{M} \rightarrow \mathbb{N}$ yields the countrywide number of teaching persons who will leave their schools at the end of school year $t$:

$$\tau_\epsilon(t, m) = \sum_{s \in \mathcal{S}} |\tau(s, t) \backslash \epsilon(s, t) \tag{3}$$

$$\epsilon(s, t) = \{p | p \in \tau(s, t) \wedge \rho(p, t) \geq 1\} \tag{4}$$

- $\tau_{\epsilon f}(t) : \mathcal{T} \times \mathcal{S} \to \mathbb{N}$ yields the countrywide number of teaching persons who will leave their schools at the end of school year $t$ and would have to be replaced with women if $\nu(t-1)$ had been 1:

$$\tau_{\epsilon f}(t,m) = \sum_{s \in \mathcal{S}} \pi(s, t-1)|\eta(s,t)| \tag{5}$$

$$\eta(s,t) = \{p | p \in \tau(s,t) \wedge \rho(p,t) < 1\} \tag{6}$$

$$\nu(t) = \tau_{\epsilon}(t) / (2\tau_{\epsilon f}(t) \tag{7}$$

For technical reasons and also for being able to derive a probability density function and a cumulative distribution function along the lines of [29] (see Appendix A), $x \in [0,1]$ had to be rescaled to $\xi = 2x - 1 \in [-1, 1]$ to produce the Figures 2–4. In the terms used in $M_p(GDS)$ the probability density function reads

$$f(x) \propto \exp[V(x)] \tag{8}$$

$$V(x) = 2(\delta - \kappa) + 2\kappa(2x-1)^2 - [2x \ln(2x) - (2-2x)\ln(2-2x)] \tag{9}$$

There are at least three reasons why the process defined in [29] is different (and much less complex) than the process defined here:

- **GDS** does not contain a process of switching between female and male. Instead an empty position is filled with a woman or with a man, and the function $\pi$ determines how many of the free positions in a certain school should be filled with a women—and as the few free positions must be filled with integer numbers of men or women something an allocation by drawing lots is performed (in the simulation model, not necessarily in the target system) to reach the desired gender ratio in this school.
- This replacement process is not defined on the state level but on the school level such that the probability of drawing a lot in favour of a women is not the same over all schools but, on the contrary, specific for each school.
- The period between two replacements in which the same person is involved (first employed, later on retired) is different for all persons, as the length of this period is a random variable with different means for men and women; as this is part of the initialisation of the model (and of new teachers) this is not reflected in $M_p(GDS)$.

Nevertheless, the resulting probability density function in **GDF** is likely to be of the same form as in Equation (9): with appropriate $\kappa$ and $\delta$ it is a good approximation for the observed frequency distributions (see Figure 2)—although the $\kappa$ and $\delta$ in Equation (9) need not necessarily be the parameters that controlled the target system. We return to this form of gender relation indicator in the context of Figure 3 (see Figure 4 and the text referring to this figure).

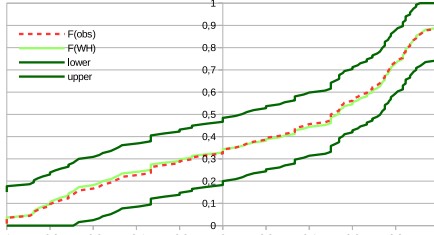 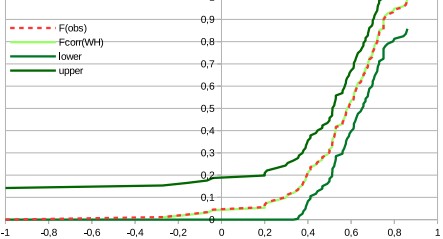

**Figure 2.** Cumulative frequency distribution (green) approximated with Equation (A5) with $\kappa = 0.7700$ and $\delta = 0.7548$ ($\theta_{KS} = 0.0156$) for school year 1950/51 (left) and $\kappa = 0.4500$ and $\delta = 0.5500$ for 1989/90 ($\theta_{KS} = 0.0073$) (right) with 80 per cent confidence bands (dark green) around the observed cumulative frequency distribution (light green).

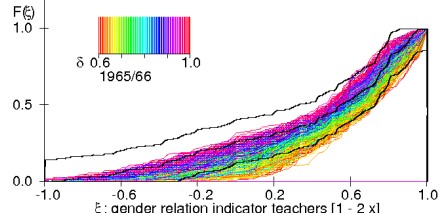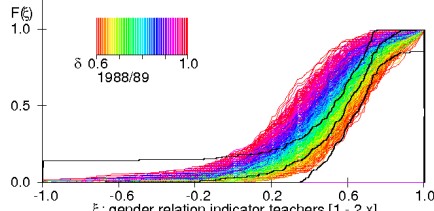

**Figure 3.** Observed cumulative frequency distribution and its 80 per cent confidence band (black) and simulated cumulative frequency distributions for varying $\delta$ (colour) between 0.6 and 1.0 and $\kappa$ between 0.0 and 1.0 (the effect of the latter is not shown ). Left: 1965/66, right: 1988/89.

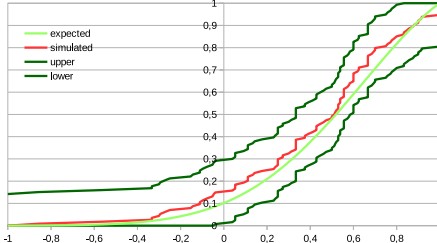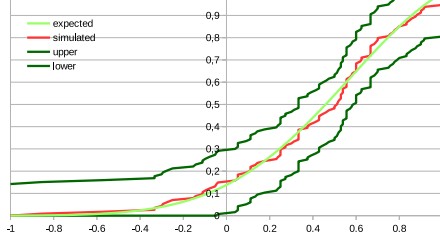

**Figure 4.** Simulated cumulative frequency distribution (red) for a run with $\kappa = 0.5701$ and $\delta = 0.9718$ and one school year—1967/68—and its 80 per cent confidence band (dark green) and the expected cumulative frequency distributions (light green) for the resulting $\kappa^* = 1.1402$ and $\delta^* = 0.4017$ (left) and with $\kappa^* = 1.1$ and $\delta^* = 1.3$ corresponding to $\kappa = 0.55$ and $\delta = 0.75$ (right).

The *full model* **M(GDS)** results from the *potential model* $\mathbf{M}_p$**(GDS)** by adding formulas to the functions mentioned in $\mathbf{M}_p$**(GDS)** which serve as axioms of **GDS**. The only axiomatic function is certainly $\pi(s,t) = \nu(t)\delta \exp(\kappa x(s,t))$ which yields a time-dependent replacement probability which could replicate the documented history (we will see that for constant $\delta$ this is not accurate enough). The other necessary functions representing the variable attributes of the MIMOSE model are straightforward, and their mechanisms are observable even when nobody cares for gender desegregation: people's ages are incremented by 1 every school year ($\alpha(p,t) = \alpha(p,t-1) + 1$)), and time till retirement is decremented by 1 every school year ($\rho(p,t) = \rho(p,t-1) - 1$), and the consequences of age and time until retirement for the staff of each school are also observable—with the restriction that the actual time of (or age at) retirement (or death or other termination) can only be estimated probabilistically, but on the grounds of some well-established theory which is also used successfully in realms such as life insurance or life annuity.

## 4. From a Structuralist Reconstruction of a Theory to a Simulation Model

The transformation of a structuralist theory reconstruction into an executable NetLogo program is also straightforward:

- Take any base set from the definition of the *potential model* (except a set of instants and a probability space) and transform it into a NetLogo `breed`. This is done by inserting
  `breed [<base set name plural> <base set name singular>]`
  for each of the base sets (except the additional $\mathcal{T}$ and $\langle \Omega, \mathcal{F}, P \rangle$ before the `to setup` procedure in the NetLogo program.

  If there is no set of instants the theory will not be about a dynamical process, and a simulation program is of no use. If the set of instants is continuous, a discretisation will always be necessary for any kind of (digital) computer simulation. So first a redesign of the theory will be necessary.
- Take any function from the definition of the *potential model* and transform it into an attribute of the respective object type, considering the domain of the respective function. This is done by inserting
  `to-report <function_name> <arguments>`

  `<...>`

```
end
```
anywhere in the NetLogo program.
- Take any axiom from the definition of the *model* of the theory and transform it into the body of a NetLogo function. This is most easily done by inserting
  ```
  set <axiom_name> <right hand side of the axiom>
  ```

  between the corresponding
  ```
  to-report <...>
    <...>
    report <axiom_name>
  end
  ```

The approach taken here has some similarity to the approach in [30,31], as the latter shows how NetLogo models can be built from ontologies and the former shows how NetLogo models can be built from the definition of *models* of a theory. Salt and Polhill called their approach "'white-boxing' models. All aspects of the model are potentially transparent, even if the code is unavailable, we have the underlying 'theory' behind the model", and this is also true for the approach described in this paper. Moreover, both approaches can be used to "combine with other models whose semantics align" [30]—a technique which resembles the building of intertheoretical relations ([12], pp. 248–250) where two theories $T_1$ and $T_2$ are linked, for instance via a term which is $T_1$-theoretical but not $T_2$-theoretical. In our example, $\rho$ could be thought of as a **GDS**-non-theoretical term, as the time until retirement is not anything whose measurement depends on the process of employing teachers and assigning them to certain schools (but needs to be measured in any theory which deals with employing, dismissing and pensioning off staff); on the other hand, at any point of time $t$ and for any employee or candidate $p$, $\rho(p, t)$—the number of years after $t$ until person $p$ retires or leaves his or her job for other reasons is not directly observable at time $t$, but there is a probability distribution for $\rho$ whose parameters can be estimated and which will, among others, depend on the gender $\beta(p)$. In the context of the current paper, both the MIMOSE and the NetLogo model include such a theory saying that $\rho(p, t_0)$—the time until retirement from the time of first employment (The age at the time of first employment is set to a random variable with mean 40 and standard deviation 7 at the time of the initialisation of the model; later on it is set to 30; both settings are somewhat arbitrary as better estimates, particularly for 1950, were not available) is normally distributed with $\mu_\rho = 30$ for men and $\mu_\rho = 15$ for women and $\sigma_\rho = 5$ for both with the additional restriction that all persons retire at an age of 65. This simple model of the ageing process is simply included in the simulation model, but it could have been programmed separately and later on "combined" in the sense of [30] with the main model. Another possible combination could be thought of with respect to the dependence of the gender relation among teachers on the gender relation among pupils. The MIMOSE model does not even model the pupils, but the NetLogo model does, although only for illustrative purposes, but a more sophisticated model of the whole process should take into account that the number of girls attending gymnasien increased considerably and much faster (from 16,817 to 31,107 or by 84.97 per cent) than the number of boys (from 23,017 to 29,020 or by 26.08 per cent) between 1950 and 1990. Hence an alternative or additional assumption would also take the changed demand side into account, for instance assuming that the change of the gender relation of teachers should reflect the change of the gender relation of pupils in some way—although this never happened as the proportion of female teachers rather decreased slightly over the years [27].

As NetLogo is a procedural language, not a declarative functional language, reporter procedures (`to-report` functions) are not the optimal solution with this toolbox; instead the typical NetLogo programmer is likely to prefer `to update-<breed-name>` (command) procedures which have to be called from within the `to go` procedure, see below. Hence Table 4 provides a translation from the definition of $M_p(GDS)$ to typical NetLogo code fragments (column 2 containing the declarations and colum 3 containing the assignments).

**Table 4.** Code lines in a NetLogo model representing the terms of the structuralist reconstruction of **GDS**.

| # | GDS Term | NetLogo Code Declarations | NetLogo Assignments |
|---|---|---|---|
| 2 | $\mathcal{T}$ | `ticks` | `tick` (at the end of the `to go` procedure) |
| 3 | $\langle \Omega, \mathcal{F}, P \rangle$ | | (hidden in NetLogo's pseudo-random number generator) |
| 4 | $\langle A, \mathcal{A} \rangle$ | | (hidden in NetLogo's pseudo-random number generator) |
| 5 | $\mathcal{M}$ | NetLogo's observer | |
| 9 | $\kappa$ | `kappa` | (set in the interface) |
| 10 | $\delta$ | `delta` | (set in the interface) |
| | | `globals [` | `to go` |
| | | | `  ask schools [ update-school ]` |
| | | | `  ask teachers [ update-teacher ]` |
| 13a | $\tau_\epsilon$ | `  teachers-to-replace` | `  set teachers-to-replace sum [ to-replace ] of schools` |
| 13b | $\tau_{\epsilon f}$ | `  teachers-to-replace-with-women` | `  set teachers-to-replace-with-women sum [ women-to-replace ] of schools` |
| 11 | $\nu$ | `  nu` | `  if teachers-to-replace-with-women > 0` |
| | | | `    [ set nu teachers-to-replace / ( 2 * teachers-to-replace-with-women ) ]` |
| 25 | $f$ and $F$ [1] | `  outfilename` | `  report-schools` |
| | | `]` | `end` |
| 6 | $\mathcal{G}$ | dispensable | |
| 12 | $\sigma$ | dispensable | |
| 7 | $\mathcal{S}$ | `breed [ schools school ]` | |
| | | `schools own [` | |
| 13 | $\tau$ | `  teacher-list` | |
| | | | `to update-school` |
| 14 | $x$ | `  sex-ratio` | `  set sex-ratio ( count teacher-list with [ color = red ] ) / count teacher-list` |
| 15 | $\lambda$ | `  sex-ratio-list` | `  set sex-ratio-list lput sex-ratio sex-ratio-list` |
| 16 | $\pi$ | `  prob` | `  set prob nu * exp ( kappa * sex-ratio )` |
| 17 | $r$ | `  random-float 1.0` | |
| 23a | $n_\epsilon$ | `  to-replace` | `  set to-replace count teacher-list with [ will-retire? ]` |
| 23b | $n_{\epsilon f}$ | `  women-to-replace` | `  set women-to-replace to-replace * prob / nu` |
| | | | `end` |
| | | `]` | |

**Table 4.** *Cont.*

| # | GDS Term | NetLogo Code Declarations | NetLogo Assignments |
|---|---|---|---|
| 8 | $\mathcal{P}$ | `breed [ teachers teacher ]`<br>`teachers own [` | |
| 18 | $\phi$ | `  my-school` | (initialised as constant) |
| 19 | $\beta$ | `  sex` | (initialised as constant) |
| | | | `to update-teacher` |
| 20 | $\alpha$ | `  age` | `  set age age + 1` |
| 21 | $\rho$ | `  duration` | `  set duration duration - 1` |
| 22 | $\omega$ | `  will-retire?` | `  if duration < 1 or age > 64` |
| | | | `    [ set will-retire?  true ]` |
| 23, 24 | $\epsilon$ and $\mu$ [2] | | `  if duration < 0 or age > 65` |
| | | | `    [ init-new-teacher ]` |
| | | | `end` |
| | | `]` | |

[1] $f$ can be seen in NetLogo's view in a histogram plot, but all data necessary to calculate better estimates of $f$ and $F$ are output to a file listing the number of men and women for every school and every year. [2] When a teacher has to be replaced this is done "in situ", i.e., `sex`, `age` and `duration` are reset. Hence `will-retire?` is used only for calculating `to-replace`.

- Finally, use NetLogo's user interface and the `to setup` procedure to initialize all constants and variable attributes which must have taken values at simulation start time, and to fix the simulation parameters (time step size, break and stop condition) and run the program.

As the model is designed to be initialised with historical data (instead of random values distributed similar to historical data as in the MIMOSE version) a complicated `to setup` procedure is necessary (this procedure reads a map of Rhineland-Palatinate and the geographical co-ordinates of all schools that existed between 1950 and 1990 together with the teacher and pupil numbers of the school year 1950/1951 from two files); moreover output files need to be written to prepare empirical validation and particularly the calculation of the cumulated simulated frequency distributions $F^{sim}_{\kappa_i, \delta_i, j}$ to compare them to the observed frequency distributions $F^{obs}_j$. Moreover, one might want to embed the simulated schools in the historical geography from which the data stem, and to make the model even more illustrative, one might want to add girls and boys to be instructed by the simulated teachers, even when they have no part in **GDS**. From all these features, the model presented in [10] (The NetLogo model is available from https://www.comses.net/codebases/7220b8df-b820-4853-a5c8-903933b04b8a/releases/1.1.0/ together with the necessary initialisation data and an analysis program which carries out all the steps needed for Section 5) arises whose results will be presented in Section 5.

## 5. Model Results and Empirical Validation

### 5.1. Validating The Extended Model

Some of the results of a NetLogo model as described in the previous section were already published in [10]. This paper will mainly deal with the validation question and the problem of estimating or even measuring terms which were only introduced by **GDS**, namely $\kappa$ and $\delta$. The third term $\nu(t)$, which is obviously not a **GDS**-non-theoretical term, turns out to be no more than a calibrating function making sure that for $\delta = 1$ men and women have the same overall probability to be employed when a position needs to be filled. In all sensitivity analyses done with various versions of the model the value of $\kappa$ seemed to be of vanishing effect, at least in the reasonable span of $0.3 < \kappa < 0.8$, hence, it was mainly $\delta$ whose effect on simulation results and their similarity with the historical data needed to be assessed. But a closer inspection of the results of a larger number of simulation runs revealed that there is some influence of $\kappa$, too, at least in the early years of the observed period. This effect is shown in Figure 5 which gives the reduction of the variance of the Kolmogorov-Smirnov distance $\theta_{KS} = \sup_{x \in \mathbb{R}} |F^{sim}_{i,j}(x) - F^{obs}_j(x)|$ between the empirical cumulative density function of the proportion of women over all schools $F^{obs}(x)$ and the respective simulation results $F^{sim}_{i,j}(x)$ from 660 runs ($i = 1 \ldots 660$) for 39 school years ($j = 3 \ldots 41$ for the school years 1952/53 $\ldots$ 1966/67, 1967, 1967/68 $\ldots$ 1989/90 (Until 1966/67, school years lasted from Easter to Easter, from 1968/1969 on they lasted from September till July, such that school year 1966 began on 1 April 1966 and ended on 30 November of the same year and school year 1967/68 lasted from 1 December 1967 till 31 July 1968. This particularity is not reflected in the model); there are no results for the first two school years, as these distances are zero or extremely small because for the first school year the simulated data are identical with the empirical data with which they were initialised while for the second school year with $\theta_{KS} = 0.050921$ the distribution is still the same in all 660 runs).

Figure 5 shows clearly that after 1969 the influence of $\kappa$ on the simulation results vanished completely, but in some of the earlier years it dominated the influence of $\delta$. If one realises what the two parameters stand for it seems that the tendency to send women applicants to women-dominated schools was still quite strong whereas the postulate to give men and women an equal chance when they applied for a teacher career was very weak (and perhaps indeed a violation of the constitutional equal opportunity postulate).

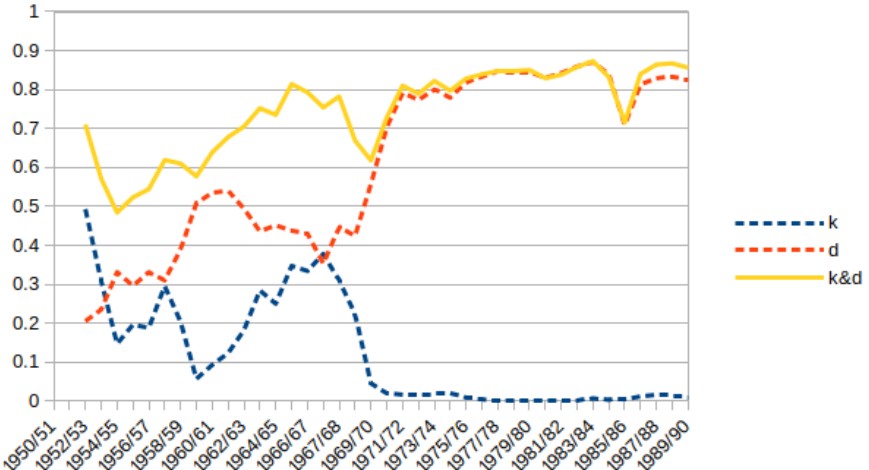

**Figure 5.** Sensitivity analysis with respect to the parameters $\kappa$ and $\delta$ ($R^2$ from regressions of $\theta_{KS}$ on $\kappa, \kappa^2$ (k), $\delta, \delta^2$ (d) and $\kappa, \kappa^2, \delta, \delta^2, \kappa\delta$ (k&d).

Taking into account that validity is a complicated concept we first discuss various aspects of this concept. Zeigler [32,33] distinguished replicative, predictive and structural validity, very recently [34] (see also [35]) introduced a similar distinction into input validation, process validation and output validation. While Zeigler's structural validation is more or less the same as Tesfatsion's and Gräbner's process validation, Zeigler's replicative and predictive validation aspects resemble Tesfatsion's and Gräbner's descriptive output validation and predictive output validation, respectively. But Tesfatsion introduced a validation aspect that was not considered by Zeigler, namely the input validation: "Are the exogenous inputs for the model empirically meaningful and appropriate for the purpose at hand? Examples of exogenous model inputs include functional forms, random shock realizations, data-based parameter estimates, and/or parameter values imported from other studies." Not included in this enumeration is the initialisation from historical data—which is the case in our example: The simulation starts with all schools that existed in 1950 and with teacher numbers as historically recorded; the annual average increase in countrywide teacher numbers is also included in the NetLogo model (although not in the considerations in Section 4), the distribution of the duration of the period of sevice by male and female teachers is estimated from historical data (although the source is not overly reliable); $\kappa$ did not seem to be a sensitive parameter, and $\delta = 1$ in the original MIMOSE version was taken from the employer's constitutional obligation to grant men and women the same chance to be employed. Hence it is the process validity and the predictive output validity ("How well are model-generated outputs able to forecast distributions ...for sample data withheld from model identfication ...")—which is mainly at stake in the current context.

*5.2. Measuring* **GDS***-Theoretical Terms*

In the original paper [7] as well as in [9] the predictive output validation was done only for $\delta = 1$ and only by a visual comparison of historical and simulated histograms showing the distribution of the percentage of female teachers over all schools over the whole period from 1950/51 until 1989/90. This comparison gave the impression that the match between the stacked histograms was satisfactory. A more detailed analysis in [6], however, showed that for most of the period under observation the Kolmogorov distance between the historical distributions and the simulated distributions was smaller for $\delta < 1.0$—which means a violation of the equal opportunities principle for the whole period. An even more detailed analysis yielded different $\kappa$ and $\delta$ values for different school years leading to good approximations of the simulated cumulated density functions $F^{sim}_{\kappa_i,\delta_i,j}(x)$ to the observed functions $F^{obs}_j(x)$. The goodness of approximation is again measured with the Kolmogorov-Smirnov distance $\theta_{KS} = \sup_{x \in \mathbb{R}} |F^{sim}_{\kappa_i,\delta_i,j}(x) - F^{obs}_j(x)|$ (runs are indexed with $i = 1 \ldots 660$, school years are indexed with

$j = 1 \dots 41$). (All these calculations were done with the help of the C program koed which is available from https://www.comses.net/codebases/7220b8df-b820-4853-a5c8-903933b04b8a/releases/1.1.0/, too) The simulation results cannot be used for point estimations of $\kappa$ and $\delta$ because quite different combinations of these parameters yield values for $\theta_{KS}$ which are small enough to be insignificant.

This is why Figure 6 gives only boxplots for $\kappa$, $\delta$ and $\theta_{KS}$ for those 50 runs and the school years which showed the smallest value for $\theta_{KS}$. As a consequence of the relative small influence of $\kappa$ on $\theta_{KS}$ (see Figure 5) the boxplot for $\kappa$ shows very large ranges and quartile differences for most of the years; one can, however, see that in early years $\kappa$ values around 0.75 prevailed whereas during the final 15 years $\kappa$ was only about 0.4. The series of $\delta$ shows high values about 0.95 from the mid-1950s until the mid-1960s, a slow decrease to 0.85 during the second half of the 1960s, stability for the next one and a half decades and from 1983 till 1986 a steep decrease to less than 0.7. The selected 50 runs for each school year are sufficiently similar to the observed data, as $\theta_{KS}$ is always below 0.12. This means that, according to the Kolmogorov-Smirnov two-sample test, the null hypothesis $|F^{sim}_{\kappa_i,\delta_i,j}(x) = F^{obs}_j(x)|$ does not need to be rejected for all these runs which were used to find estimates for $\kappa$ and $\delta$, and that all these runs yielded cumulative distribution functions lying well within the 80 per cent confidence band [36], ([37], chapter 7), the half width of this band being 0.14209 for a sample size of 114. Examples for the 80 per cent confidence band for two years are shown in Figure 3 which uses the conversion from $x$ to $\xi$ to be compatible with the relation between switching function and probability density function mentioned by [29]. There is no simulation run whose simulated distribution is within the 80 per cent confidence band around the observed distribution for all years; only run 441 with $\kappa = 0.829$ and $\delta = 0.831$ produced distributions for all years up to 1977/78 within this confidence band (which is clearly in line with what can be observed in Figure 3).

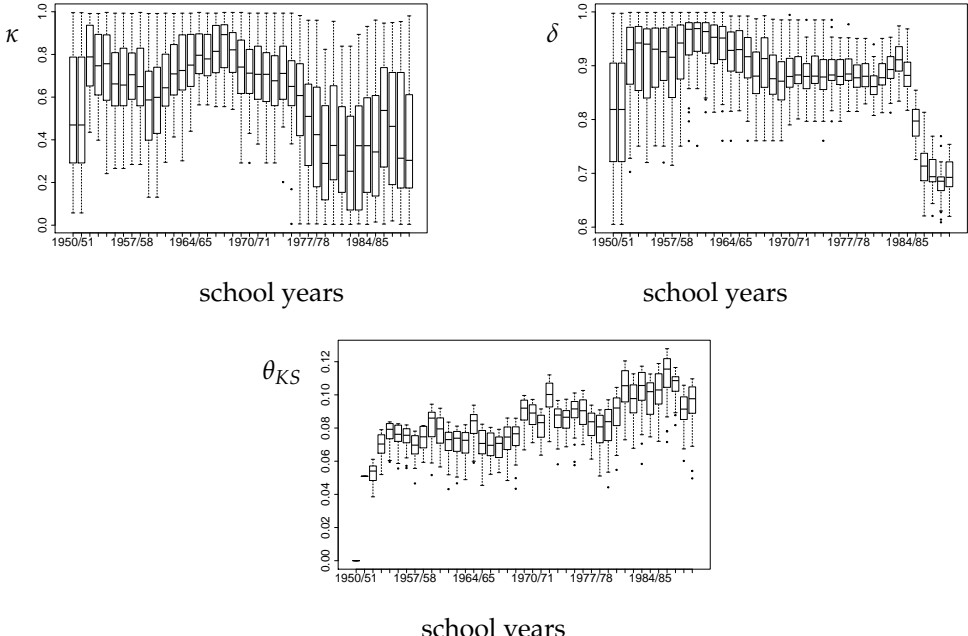

**Figure 6.** Left top: boxplot of the distribution of $\kappa$ for the 50 best runs (out of 660 runs), right top: boxplot of the distribution of $\delta$ for the 50 best runs (out of 660 runs), bottom: boxplot of the distribution of $\theta_{KS}$ for the 50 best runs (out of 660 runs).

From Figure 3, it is easily seen that for the school year 1965/66 simulation runs with high $\delta$ (blue and violet) match the observed distribution whereas for the school year 1988/89 this holds for $\delta \approx 0.75$ (green, yellow and orange).

Additionally, Figure 4 shows (for one run with $\kappa = 0.5701$ and $\delta = 0.9718$ and one school year—1967/68—) that the probability function calculated from the resulting $\kappa^* = -1.1402$ and

$\delta^* = 1.5419$ again lies well within the 80 per cent confidence band around the simulated cumulated frequency function. That $\kappa^* = -1.1$ and $\delta^* = 1.3$ leads to an even better approximation between the two curves shows that there are still differences between the approximative analytical solution following the concept of [29] and the stochastic model realised in the NetLogo model—although these differences are not statistically significant. This difference can be explained as a consequence of the fact that the process considered here is not one of individuals changing their gender according to the prevailing overall majority in their respective schools but of individuals replacing each other both locally and after a long period.

## 6. Conclusions

This paper showed several aspects of modelling social phenomena on several levels, aspects which are typical in multilevel and agent-based modelling:

- the task of verification which can be supported by a structuralist reconstruction of a mental model or an available simulation model (Section 3),
- the task of replication which can also be supported by such a reconstruction and by building a new model, perhaps with entirely different tools, on the grounds of the structuralist axiomatisation (Sections 3 and 4),
- the task of validation (Subsection 5.1) as the structuralist reconstruction makes clear which of the simulation model terms represent real-world terms which can be observed without using the theory whose *potential and partial models* are defined in this reconstructions—here **GDS**-non-theoretical terms—and which of these terms can only be made measurable as **GDS**-theoretical terms,
- finally the task of measuring **GDS**-theoretical terms with the help of simulation output by finding out which values of these terms lead to simulation runs whose output fits measurements of **GDS**-non-theoretical terms best (Subsection 5.2).

The latter task can be compared to measuring natural constants such as Newton's gravitational constants or Coulomb's electrostatic constant whose values could be found out only after Newton's and Coulomb's laws stating the proportionality of a force to the product of two masses or charges, divided by the square of the distance, had been established. (Another task of a similar kind is deriving device constants from comparable laws, see [6] mentioning the spring constant using Hooke's law or [38] mentioning the device constant of a crossbow using the relation between the angle at which the arrow was shot, the maximum height of the arrow's way through the air, and the maximum length of the arrow's flight)

Hence, the intended application of **GDS** to the data about the staff of German schools in the second half of the past century led to a simulation model (and some additional analysis software calculating the cumulative distribution functions of the simulation results) which allows for stating the relation between

- the distribution $Fx$ of gender proportions $x$ in the teaching staff of school and
- two traits of officials responsible for employing teachers, namely

  - to select men and women with a certain preference for one gender ($\delta \neq 1$) or no such preference leading to equal opportunities ($\delta = 1$) and
  - to send women preferably to schools dominated by women and not to send men to such schools ($\kappa$ high) or to send persons regardless of their gender to schools of any degree of gender relations ($\kappa$ low).

The simulation results showed that $F(x)$ or, to be more precise, the "measure of validation" ([39], p. 163), namely the variance of $\theta_{KS} = \sup_{x \in \mathbb{R}} |F_{i,j}^{sim}(x) - F_j^{obs}(x)|$ is so well explained by $\kappa$ and $\delta$ that at least an estimation of the historical values of these two **GDS**-theoretical terms became possible which revealed that during the first two decades of the documented period the strategy applied was

to segregate men and women (and also boys and girls), leaving $f(x)$ a bimodal frequency density function, whereas in the third decade the influence of $\kappa$ and $\kappa$ itself decreased and $f(x)$ became unimodal, $\delta$ still being high but still below the equal opportunities level of 1.0, whereas the fourth decade saw a decrease of both values, such that the allocation of applicants to schools became more and more gender-independent, and equal opportunities were not achieved (Kraul, M.; Wirrer, R. ([27], p. 322) even state that "*the primacy which female teachers used to have in girls schools was given up in the process of establishing coeducation*, but no kind of 'compensation' was achieved" (italics in the original, my translation) . As this happened only during the last few years of the observed period and no further data could be collected, the allocation strategies after 1990 cannot be evaluated with the help of **GDS**.

But anyway, the results of this paper seem to encourage to connect agent-based and multilevel simulation models to empirical data and particularly show ways to connect global parameters to emergent phenomena such as distributions of features of lower level entities, and it also shows that "the requirement of an explicit formulation of the hypotheses with the indication of the underlying assumptions is perfectly realisable when one uses the metatheoretical tool of the structuralist metatheory" ([14], my translation).

**Funding:** Part of this paper, particluarly Section 3 originates in the research project "Micro and multilevel modelling and simulation software development (MIMOSE)" directed by Michael Möhring and the author between 1988 and 1992 and funded by the Deutsche Forschungsgemeinschaft under grant no. Tr 225/3–1 and –2. The research from which the data stem which lead to the results of Section 5 and allowed first steps into designing the model in Section 3 received funding from Deutsche Forschungsgemeinschaft between 1991 and 1995 under grant agreement no. KR 960/5–1 and –2 ('Einführung und Auswirkung der Koedukation. Eine Untersuchung an ausgewählten Gymnasien des Landes Rheinland-Pfalz'/'The introduction of co-education. A study of educational history at selected grammar schools in the German State of Rhineland-Palatinate', directed by Margret Kraul).

**Acknowledgments:** The diagram in Figure 1 goes back to discussions with Martin Ihrig and was slightly changed for the purposes of this paper. The paper has profited from the comments of three reviewers which are gratefully appreciated.

**Conflicts of Interest:** The author declares no conflict of interest. The founding sponsors had no role in the design of the study; in the collection, analyses, or interpretation of data; in the writing of the manuscript, and in the decision to publish the results.

## Appendix A. An Alternative Measure for the Gender Relation

With respect to $x$, another form of gender relation indicator (That we use both $x$ and $\xi$ in this paper is mainly due to the fact that in earlier publications the frequency density function $f^*(\xi)$ was used and that the computer program used to analyse observed and simulated data also used $f^*(\xi)$ instead of $f(x)$) is possible when one goes back to ([29], pp. 40–44) who used the functions

$$p_{12} \;=\; \nu \exp(\delta^* + \kappa^* \xi) \tag{A1}$$
$$p_{21} \;=\; \nu \exp[-(\delta^* + \kappa^* \xi)] \tag{A2}$$

in a model of a population consisting of individuals switching between two opinions ("yes" and "no") where the probability of such a switch depended on the current majority—which was described by the variable

$$\xi = \frac{n_{yes} - n_{no}}{n_{yes} + n_{no}}, -1 \le \xi \le +1 \tag{A3}$$

In [29] and showed that the resulting probability density function was of the form

$$f^*(\xi) \;\propto\; \exp[U(\xi)] \tag{A4}$$
$$U(\xi) \;=\; 2\delta^* + \kappa^* \xi^2 - [(1+\xi)\ln(1+\xi) - (1-\xi)ln(1-\xi)] \tag{A5}$$

With $\xi = 2x - 1$ ($\xi = 1$ is equaivalent to $x = 0$ or no women, $\xi = -1$ is equivalent to $x = 1$ or no men!) this results in $\delta^* = \delta - \kappa$ and $\kappa^* = 2\kappa$.

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
