# Peer review of "Axiomatisation and Simulation"

_information, doi:10.3390/info10020053_

Round 1

Reviewer 1 Report

This paper presents the MIMOSE approach to describe a relatively simple ABM, calibrated on statistical data. The model, as declared by the author, had already been published in ref. [8,9]. While this reviewer thinks that it deserves publication, the reviewer is left with the impression is that the main ideas in the paper could be presented more clearly, and that some of the details, of some of the ideas thrown in but not developed (geography for example) could be moved in an appendix or simply skipped.

For this reviewer, the main contributions are
 - the approach as a whole (Fig. 1) which is indeed explained in some detail. It still feels to this reader as if containing too many elements. Maybe it could be separated in levels of increasing detail. Other simpler suggestions are added in the annotated file that follows.
  - the clear representation of interaction between theories (10:312 and following.)

The refutation of Axelrod's "third way," announced in the introduction, is actually not carried on convincingly in the text.

Formula 2 is thrown in too early and without justification. Maybe it is discussed in the other papers or customary, but I wonder why the exponential form, which is not by itself limited to the interval [0,1]. Would any increasing function do?

More detailed comments in the attached annotated pdf.

Author Response

Dear colleague, thanks for your helpful comments. I have revised the manuscript and followed most of your suggestions, see below.

Point 1: This paper presents the MIMOSE approach to describe a relatively simple ABM, calibrated on statistical data. The model, as declared by the author, had already been published in ref. [8,9]. While this reviewer thinks that it deserves publication, the reviewer is left with the impression is that the main ideas in the paper could be presented more clearly, and that some of the details, of some of the ideas thrown in but not developed (geography for example) could be moved in an appendix or simply skipped.

My answer: I have tried to present the main ideas more clearly, mainly by inserting additional text in the introduction and in section 2, particularly to make clear that the main aim of the paper is to show a "stucturalist reconstruction" of an old model and the advantage of such a reconstruction for the purpose of replication of the model in another toolbox. Some details were moved to the appendix.

Point 2: For this reviewer, the main contributions are
 - the approach as a whole (Fig. 1) which is indeed explained in some detail. It still feels to this reader as if containing too many elements. Maybe it could be separated in levels of increasing detail.

My answer: As some details were moved to the appendix, I hope that this problem is solved. Moreover additional text was inserted to connect the different elements better.

Point 3: Other simpler suggestions are added in the annotated file that follows.

My answer: Thanks for these; I left the word "desk research" because "Desk" was mentioned in Figure 1, and I did not want to change too much in this diagram. I have also abstained from numbering the sequences, as only few of them are explicitly mentioned, and these are sufficiently addressed (at least when one can see them in colour). All the other comments were strictly followed, as the new version (with new text in red and discarded text crossed out) shows. I was wondering why you could not access the NetLogo model at Comses --- it is there; perhaps it was unaccessible because it is still awaiting reviewer feedback, but the website says that one can "Use this link to share this release privately with others. Anyone with this URL will be able to access this release." Here is the link again, in case it was misspelled in the first version:  https://www.comses.net/codebase-release/01c2d4d6-3b5a-4722-82da-fb7cc66e4dd8/

My answer: Thanks for these; I left the word "desk research" because "Desk" was mentioned in Figure 1, and I did not want to change too much in this diagram. I have also abstained from numbering the sequences, as only few of them are explicitly mentioned, and these are sufficiently addressed (at least when one can see them in colour). All the other comments were strictly followed, as the new version (with new text in red and discarded text crossed out) shows. I was wondering why you could not access the NetLogo model at Comses --- it is there; perhaps it is because it is still awaiting reviewer feedback, but the website says that one can "Use this link to share this release privately with others. Anyone with this URL will be able to                        access this release." Here is the link again, in case it was misspelled in the first version:  https://www.comses.net/codebase-release/01c2d4d6-3b5a-4722-82da-fb7cc66e4dd8/

Point 4:   - the clear representation of interaction between theories (10:312 and following.)

My answer: Thanks for the compliment.

Point 5: The refutation of Axelrod's "third way," announced in the introduction, is actually not carried on convincingly in the text.

My answer: I hope the additional text is now more convincing.

Point 6: Formula 2 is thrown in too early and without justification. Maybe it is discussed in the other papers or customary, but I wonder why the exponential form, which is not by itself limited to the interval [0,1]. Would any increasing function do?

My answer: The use of this formula is now justified; the main reason is that this kind of function allows for bimodal distributions and that it has an approximate closed solution for the emerging probability density function. Not all increasing functions would meet both requirements.

Reviewer 2 Report

This paper provides a very interesting description of the development of an exciting and complicated simulation model. I feel that the literature will profit a lot from this documentation of how a leading scholar developed a model and his methodological reflections about this process. Furthermore, I am a great fan of one of the paper’s general messages (“refuting the idea that simulation is a “third way of doing science” [12] or “a distinctively new kind of scientific method”). This is an important contribution to the literature.

My main concern is that the paper lacks focus, in that it is not clear what the reader is supposed to learn from it. To be sure, this paper is very rich, but I recommend stating very clearly at the beginning of the paper what the paper demonstrates and then target the analyses at this aspect. In the current version, various interesting contributions are mentioned in the intro and the conclusion section. For instance, it is mentioned in the intro that the paper aims at “refuting the idea that simulation is a “third way of doing science” [12] or “a distinctively new kind of scientific method”. This is great, but this aim is not explicitly addressed in the paper. In order to do this, it would be necessary to review the debate about this issue and to explain to the reader why the presented information refutes this notion. Alternatively, the paper could also be focused on the findings of the analyses that are summarized in the conclusion section (e.g. officials send tend “to send women preferably to schools dominated by women and not to send men to such schools”). If this is turned into the paper’s focus, however, it should be mentioned in the introduction and the related literature should be discussed. In addition, the title should be adjusted.

I recommend to avoid (or better define early on in the paper) the concept of “logical reconstructions of theories” (see introduction), as the term “construction” has been used in many different, and often problematic, ways. I feel that what the author aims to express is that he translated a set of assumptions into a formal language and then used some rigorous calculus to identify the implications of these assumptions. I do not see what this has in common with a process of construction. 

Likewise, I recommend more careful use of the term “logic”. In Section 2, for instance, logic seems to refer to a methodological approach to science or a “research architecture”. This deviates from the more common usage of this term, as the study of inference. I recommend to avoid this term or to provide a clear definition.

Author Response

Point 1:This paper provides a very interesting description of the development of an exciting and complicated simulation model. I feel that the literature will profit a lot from this documentation of how a leading scholar developed a model and his methodological reflections about this process. Furthermore, I am a great fan of one of the paper’s general messages (“refuting the idea that simulation is a “third way of doing science” [12] or “a distinctively new kind of scientific method”). This is an important contribution to the literature.

My answer: Thanks for the compliments.

Point 2: My main concern is that the paper lacks focus, in that it is not clear what the reader is supposed to learn from it. To be sure, this paper is very rich, but I recommend stating very clearly at the beginning of the paper what the paper demonstrates and then target the analyses at this aspect. In the current version, various interesting contributions are mentioned in the intro and the conclusion section.

My answer: I have inserted several sentences in the introduction and in section 2 which hopefully address this concern satisfactorily.

Point 3: For instance, it is mentioned in the intro that the paper aims at “refuting the idea that simulation is a “third way of doing science” [12] or “a distinctively new kind of scientific method”. This is great, but this aim is not explicitly addressed in the paper. In order to do this, it would be necessary to review the debate about this issue and to explain to the reader why the presented information refutes this notion.

My answer: I have added some discussion about "ways of doing science" which hopefully make clear that the difference between different formal descriptions of theories and models does nit justify to call simulation something entirely different from earlier ways of doing scoience.

Point 4: Alternatively, the paper could also be focused on the findings of the analyses that are summarized in the conclusion section (e.g. officials send tend “to send women preferably to schools dominated by women and not to send men to such schools”). If this is turned into the paper’s focus, however, it should be mentioned in the introduction and the related literature should be discussed. In addition, the title should be adjusted.

My answer: This is not the paper's focus, the use case is mainly taken as an example which I find appropriate to show that simulation can profit from structuralist reconstruction.

Point 5: I recommend to avoid (or better define early on in the paper) the concept of “logical reconstructions of theories” (see introduction), as the term “construction” has been used in many different, and often problematic, ways. I feel that what the author aims to express is that he translated a set of assumptions into a formal language and then used some rigorous calculus to identify the implications of these assumptions. I do not see what this has in common with a process of construction.

My answer: The founders of the "structuralist view on theories" or the "non-statement view" have always used the term "logical reconstruction" or "stucturalist reconstruction" for what they have been doing. This is now explained in the text and in a footnote. Using another word not containing "construct" would certainly be misleading and hide the connection to the work of Balzer, Sneed and Moulines.

p, li { white-space: pre-wrap; }

Point 6: Likewise, I recommend more careful use of the term “logic”. In Section 2, for instance, logic seems to refer to a methodological approach to science or a “research architecture”. This deviates from the more common usage of this term, as the study of inference. I recommend to avoid this term or to provide a clear definition.

My answer: The word "logic" or "logical" appeared in three contexts: "logical reconstruction of", and here it is rather a quotation from titles of similar papers (which is now mentioned explicitly in a footnote), "logic of the basic model of the modelling and simulation process" where this more or less a quotation (from Gilbert & Troitzsch 2005, p. 17, and Hassan et al. 2010 as well as several others using similar diagrams), and in the headline of section 2, where "logic" was mistakenly used and is now replaced with "role" (as it should have been from the very beginning).

Reviewer 3 Report

This manuscript deals with a methodological discussion of agent-based simulations. Through an analysis of a gender desegregation process in German schools in four decades, it shows the importance of connecting simulation models and a structuralist reconstruction of theories. Despite the author's discussion whose essence was repeated in his previous works is useful for the agent-based simulation researchers, I assess that this manuscript should be required the substantial and significant revision in the following point.

If this manuscript is regarded as a refereed paper, both the originality of the paper and the contribution of the academic field are not clear. Which would the Author like to stress either the importance of agent-based simulations or a gender desegregation process? If the former, the way is a difficult path, because many papers and researchers have already stressed that this methodology can be useful as the third way of science. Although I positively assess Figure 1, a scientific journal seems not to favor a metaphysical discussion. If the latter is right, the structure of the manuscript should be revised drastically and the difference from the previous papers should be clearly pointed out.

Author Response

Point 1:This manuscript deals with a methodological discussion of agent-based simulations. Through an analysis of a gender desegregation process in German schools in four decades, it shows the importance of connecting simulation models and a structuralist reconstruction of theories. Despite the author's discussion whose essence was repeated in his previous works is useful for the agent-based simulation researchers, I assess that this manuscript should be required the substantial and significant revision in the following point.

My answer: Following the comments of the other two reviewers I have added some text and moved some text to other places which hopefully is the significant revision which you demand.

Point 2: If this manuscript is regarded as a refereed paper, both the originality of the paper and the contribution of the academic field are not clear.

My answer: I hope the revised version meets your requirements better.

Point 3: Which would the Author like to stress either the importance of agent-based simulations or a gender desegregation process?

My answer: In fact, neither. I have made clear quite early in the revised version that the main concern of this paper was to show that the "logical reconstruction" or "structuralist reconstruction" of the protagonists of the "non-statement view" can be helpful for a formal design and for the replication of simulation models, and this not only in physics (where many attempts can be found in the literature) but also in the social sciences (where such attempts are still scarce).

Point 4: If the former, the way is a difficult path, because many papers and researchers have already stressed that this methodology can be useful as the third way of science.

My answer: The use case is an agent-based model, of which plenty have indeed been published, but only few have been described with this kind of formalism.

Point 5: Although I positively assess Figure 1, a scientific journal seems not to favor a metaphysical discussion. If the latter is right,

My answer: I do not think that a "structural reconstruction" is a metaphysical discussion.

Point 6: the structure of the manuscript should be revised drastically and the difference from the previous papers should be clearly pointed out.

My answer: I hope the revised version meets your standards better. There are two kinds of previous papers: some are also structuralist reconstructions of simulation model, but of models with entirely different target systems; two others deal with the same use case, but without the high formalisation adopted here. I guess that both kinds of difference are clear enough, as the first group is mentioned in the first half of the first sentence of the paper as "publications on the same topic", namely axiomatisation and simulation, whereas the second group is mentioned in the second half of the first sentence as use-case related.

Round 2

Reviewer 1 Report

No further comments.

Reviewer 3 Report

The revised version has been substantially improved. Although I concern the originality of the manuscript as I pointed out in the first round, I admit this type of the manuscript which is positively assessed by the other reviewers.